# ENRICHING KNOWLEDGE DISTILLATION WITH INTRA-CLASS CONTRASTIVE LEARNING

## ABSTRACT

Since the advent of knowledge distillation, much research has focused on how the soft labels generated by the teacher model can be utilized effectively. Allen-Zhu & Li (2020) points out that the implicit knowledge within soft labels originates from the multi-view structure present in the data. Feature variations within samples of the same class allow the student model to generalize better by learning diverse representations. However, in existing distillation methods, teacher models predominantly adhere to ground-truth labels as targets, without considering the diverse representations within the same class. Therefore, we propose incorporating an intra-class contrastive loss during teacher training to enrich the intra-class information contained in soft labels. In practice, we find that intra-class loss causes instability in training and slows convergence. To mitigate these issues, margin loss is integrated into intra-class contrastive learning to improve the training stability and convergence speed. Simultaneously, we theoretically analyze the impact of this loss on the intra-class distances and inter-class distances. It has been proved that the intra-class contrastive loss can enrich the intra-class diversity. Experimental results demonstrate the effectiveness of the proposed method.

## 1 INTRODUCTION

Knowledge distillation (KD) is a technique in deep learning where a smaller model is trained to replicate the behavior of a larger, more complex model Hinton et al. (2015); Ji et al. (2023). This approach is valuable for compressing large models Kim et al. (2018); Jin et al. (2021); Gu et al. (2023), transfer learning Vapnik et al. (2015); Noroozi et al. (2018) and enhancing the performance of smaller models Buciluă et al. (2006); Romero et al. (2014). Therefore, KD has gained popularity across various tasks, such as image classification Mobahi et al. (2020); Yuan et al. (2020), natural language processing Rashid et al. (2020); Yang et al. (2020), and multimodal learning Dai et al. (2022).

Soft labels play a crucial role in knowledge distillation by offering more comprehensive information regarding the data distribution Menon et al. (2021); Zhou & Song (2021). They encapsulate the knowledge of the teacher model, encompassing not only the relative probabilities of different classes but also the intra-class variances, which are not reflected in the hard labels. Allen-Zhu & Li (2020) proposes the multi-view data assumption, which is validated by real-world datasets, and demonstrates that students can learn features of other classes present in the soft labels. These points also constitute the core hypothesis of this paper, that soft labels express the variances among samples of the same class, which is absent in ground-truth labels.

In fact, existing methods for training teacher models can be roughly divided into two categories. One targets the ground-truth (possibly with regularization) Tian et al. (2019); Chen et al. (2021a), and the other involves joint training of teacher and student models Chen et al. (2020); Xu et al.; Neitz et al. (2020); Yuan et al. (2021), where the teacher's soft labels could be adjusted based on feedback from the student. These two methods are neither designed for intra-class diversity nor provide parameters to control intra-class variation. The soft labels generated by the teacher model may fail to capture a significant amount of valuable intra-class information.

Therefore, we propose incorporating an intra-class contrastive loss as an auxiliary loss during teacher training. This can enrich the intra-class information within soft labels, preventing the soft labels from being overly similar to the ground truth due to the model's strong fitting capability. Similar to

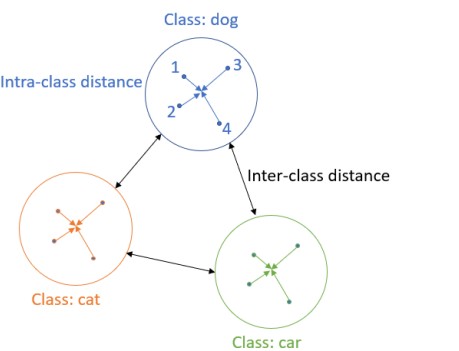 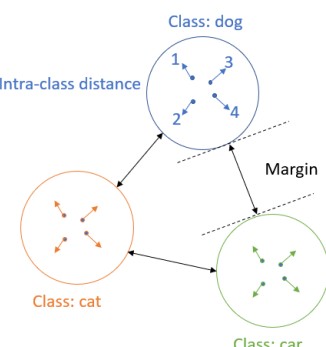

Figure 1: In most distillation methods, the teacher model is trained such that samples from different classes are separated from each other while samples from the same class are brought closer together, as illustrated in the left. However, as the intra-class distance decreases, the intra-class information within the soft labels may be lost. The proposed intra-class contrastive loss encourages an appropriate increase in intra-class distance. We also introduce a margin to ensure the stability of inter-class distances, as shown on the right.

conventional contrastive learning Sohn (2016a); Oord et al. (2018), we employ augmented samples as positive samples and other samples from the same class as negative samples. This enables the teacher model to learn intra-class variability. However, the intra-class contrastive loss may lead to two issues: the potential for mode collapse and low convergence speed. To address these issues, we integrate the concept of margin loss into intra-class contrastive learning. Besides, we implemented a pipeline-based caching mechanism to reduce memory usage and improve training stability under GPU memory constraints.

Furthermore, we analyzed the impact of the intra-class contrastive loss on the intra-class distances and inter-class distances. Initially, we demonstrated a quantitative relationship between intra-class contrastive loss and the distances within and between classes. Moreover, for the teacher model targeted at the proposed loss, we proved that the model's intra-class diversity and the proportion of intra-class contrastive loss satisfy specific constraints. As the proportion of intra-class contrastive loss increases, so does the intra-class diversity of the teacher model. This further substantiates the effectiveness of our proposed intra-class contrastive learning approach. Our main contributions are summarized as follows:

- We propose intra-class contrastive Learning to enrich the diversity of intra-class samples and introduce margin loss to enhance training stability and convergence speed.

- It is theoretically proved that the proposed inter-class contrastive loss contributes to increasing intra-class diversity, and the weights can be adjusted to control this diversity. The effectiveness and stability of our method are demonstrated.

- Experimental results on benchmark datasets also validate the effectiveness of the proposed method.

## 2 RELATED WORK

**Knowledge Distillation** has gained significant attention Buciluă et al. (2006); Hinton et al. (2015) in the field of deep learning. It was initially proposed for model compression Kim et al. (2018); Jin et al. (2021), and later widely adopted for knowledge transfer Vapnik et al. (2015); Zagoruyko & Komodakis (2016), or as a trick to enhance performance Guo et al. (2023); Jin et al. (2023). In traditional knowledge distillation, one teacher model teach one student model. Self-distillation Zhang et al. (2019); Lee et al. (2019) is a variant of distillation where a single model acts both as the teacher and the student. During training, a tradeoff between the ground-truth and the model's previous outputs is used as the target for subsequent training. Typically, the tradeoff parameter in self-distillation varies across epochs. In ensemble distillation You et al. (2017); Zhu et al. (2018), the outputs from

multiple teachers are integrated and used to instruct the student. In mutual learning (peer learning) Zhang et al. (2018); Chen et al. (2020), multiple models learn from each other or use the aggregated knowledge as a common teacher.

Although there are many variants of distillation, a significant focus remains on the dark knowledge hidden in soft labels. Most explanations regarding dark knowledge are empirically validated. However, theoretical analyses of this concept are diverse and subject to debate. Phuong & Lampert (2019) studied the distillation mechanism with assumption that both the teacher and the student models are linear. Similarly, Allen-Zhu & Li (2020) hypothesized that data adhere to a multi-view structure, where samples from different classes may share similar features. They demonstrated the effectiveness of soft labels, which stems from their ability to enable learning of information from other classes present in the samples. Research by Mobahi et al. (2020) analyzed the teacher model in self-distillation using Green's function, and regard self-distillation as a form of regularization. On a statistical front, Menon et al. (2021) and Zhou & Song (2021) treated the generated soft labels as posterior probabilities and posited the existence of the Bayes probability. Additionally, there are other studies analyzing soft labels from the perspective of transfer risk Ji & Zhu (2020); Hsu et al. (2021).

**Contrastive Learning** has emerged as a powerful mechanism for learning effective representations Logeswaran & Lee (2018); Oord et al. (2018); Tian et al. (2020a) by contrasting positive pairs against negative pairs Mikolov et al. (2013). This technique is primarily used in unsupervised or self-supervised learning environments where labeled data is scarce or expensive to obtain. Contrastive earning hinges on the idea that an encoder should map semantically similar (positive) samples closer in the embedding space, while semantically different (negative) samples should be farther apart Hadsell et al. (2006); Sohn (2016a). Sohn (2016b) introduces multi-class N-pair loss instead of traditional triplet loss. Wu et al. (2018) proposes instance discrimination to learn an embedding that can repell each pair of two training data. He et al. (2020) uses a dynamic dictionary with a momentum-updated encoder to efficiently handle large-scale data in contrastive learning so that more negative samples can be used. Gutmann & Hyvärinen (2010) assumes that the data originate from a distribution that can be parametrically clustered.

In addition to empirical work, there are some theoretical results in contrastive learning. Saunshi et al. (2019) analyzed the generalization error bounds for downstream classifiers with representations obtained from contrastive learning, based on the assumption of latent classes. Saunshi et al. (2019) argue that reducing the mutual information between views is beneficial to downstream classification accuracy. Tian et al. (2020b) proved that contrastive loss can help align features from positive pairs and features from different classes will uniformly distributed on the hypersphere. Parulekar et al. (2023) demonstrated that the representations learned by minimizing InfoNCE Loss, even with a limited number of negative samples, are consistent with the clusters inherent in the data. They further proved that when combined with a two-layer ReLU head, the learned representations can achieve zero downstream error on any binary classification task that preserves clustering. HaoChen et al. (2021) analyzed contrastive learning by constructing an augmentation graph and demonstrated that features obtained by minimizing spectral contrastive loss have provable accuracy guarantees under linear probe evaluation.

## 3 METHOD

In this section, we propose intra-class contrastive learning and define intra-class contrastive loss based on (n+1)-tuplet loss Sohn (2016b). Furthermore, we introduce margin loss to enhance model stability and accelerate convergence.

### 3.1 PRELIMINARY

Let $\mathcal{X}$ be the sample space, $\mathcal{Y} = \{1, 2, \ldots, c\}$ be the label space with $c$ classes. Given a sample $x \in \mathcal{X}$, $y \in \mathcal{Y}$ is the corresponding true label. Define $C(x)$ as the set of samples with the same class as $x$. In other words, $x' \in C(x)$ implies that $x'$ and $x$ belong to the same class. Define $\mathcal{F} : \mathcal{X} \to \mathcal{Y}$ as the hypothesis space. Each $f \in \mathcal{F}$ is a classifier. $f_t$ and $f_s$ are used to represent the teacher model and student model respectively. Similarly, $p_t$ and $p_s$ are used to represent the soft labels output by the teacher and student models. Let $x^+$ represent the positive sample of $x$. In this paper, all $x^+$ are

augmented versions of $x$. Let $x^-$ represent the negative sample of $x$. We employ tuplet loss as the contrastive loss function and the classical $(n+1)$-tuplet loss is defined as follows:

$$\mathcal{L}_{Tuplet} = -\log \frac{\exp(f(x)^\top f(x^+))}{\exp(f(x)^\top f(x^+)) + \sum_{j=1}^{n} \exp(f(x)^\top f(x_j^-))}. \tag{1}$$

The tuplet loss encourages reducing the distance between the positive pair $(x, x^+)$ and increasing the distance between the negative pair $(x, x^-)$.

## 3.2 INTRA-CLASS CONTRASTIVE DISTILLATION

Knowledge distillation involves teaching the student model to mimic the outputs of the teacher model. The loss function for the student model is defined as:

$$\mathcal{L}_{KD} = \alpha \mathcal{L}_{CE}(y, p_s) + (1 - \alpha)\mathcal{L}_{CE}(p_t, p_s) \tag{2}$$

where $\mathcal{L}_{CE}$ is the cross-entropy loss, $y$ is the ground-truth label, $p_t$ is the soft distribution predicted by the teacher, $p_s$ is the hard distribution predicted by the student, and $\alpha$ is a weighting factor. In fact, the two parts of the loss in Equation 2 can be combined into a single term:

$$\mathcal{L}_{KD} = \mathcal{L}_{CE}(q, p_s), \tag{3}$$

where $q = \alpha y + (1 - \alpha)p_t$. Equation 3 provides a more intuitive demonstration of the student model's objective. From this, we can observe that distillation essentially involves making the student model approximate the soft label $q$. The dark knowledge in the soft label $q$ originates from the teacher's output $p_t$. Existing distillation methods mostly focus on how the student aligns with the teacher, paying less attention to the learning objectives of the teacher model. In most distillation approaches, the teacher typically learns from real labels (with regularization). If the network has strong fitting capability, it may also result in the teacher's soft labels being very similar to the real labels. This can potentially lead to the reduction of intra-class information within soft labels, consequently resulting in performance degradation.

Therefore, we propose incorporating intra-class contrastive learning into teacher training to extract intra-class information. Traditional contrastive learning aims to maximize inter-class distance and minimize intra-class distance to learn discriminative and robust representations. Intra-class contrastive learning encourages the teacher model to learn embeddings where samples from the same class are dispersed appropriately, while still being distinguishable within their respective classes, thus enriching the soft label information in knowledge distillation.

In detail, we adopt contrastive loss function similar to 1. For an anchor sample $x$, we use its augmented view as the positive sample $x^+$ and other samples from the same class as negative samples $x^-$. The intra-class loss function is defined as:

$$\mathcal{L}_{Intra} = -\log \frac{\exp(f(x)^\top f(x^+))}{\exp(f(x)^\top f(x^+)) + \sum_{k=1}^{m} \exp(f(x)^\top f(x_k^-))}. \tag{4}$$

where $m$ is the number of negative samples, $\{x_k^-\}$ is the set of negative samples. The primary distinction between the intra-class loss and the classical tuplet loss lies in the selection of negative samples. The former selects negative samples with the same class as $x$, while the latter mostly chooses negative samples with classes different from $x$. Then, the total objective function of the teacher model is defined as:

$$\mathcal{L}_{Teacher} = \mathcal{L}_{CE}(y, p_t) + \lambda \mathcal{L}_{Intra}. \tag{5}$$

Here, $\lambda > 0$ represents the weight of the intra-class loss. In the entire loss function, cross-entropy loss constitutes the major component. This is quite evident as the teacher model needs to first be able to distinguish between class categories before enhancing intra-class diversity. From a clustering

---

**Algorithm 1** SGD for Margin-Based Intra-Class Contrastive Distillation

---

    **Input:** Training data $\{(x_i, y_i)\}$, learning rate $\eta$, margin threshold $\delta$, balance parameter $\lambda$ and maximum iteration T

    **Output:** Trained parameters $\theta$

1: Initialize parameters $\theta$ of the teacher model;
2: **for** t=1 **to** T **do**
3:     **for** each batch $\{x_i, y_i\}$ from the training data **do**
4:         Evaluate $p_{t_i} = \text{Softmax}(f_\theta(x_i))$ for each $x_i$ in the batch;
5:         Determine $\rho_{x_i} = p_{x_i}^{y_i} - \max_{j \in \mathcal{Y} \setminus \{y_i\}} p_{x_i}^j$ for each $x_i$;
6:         Calculate $\mathcal{L}_{Intra}$ as 7 and the total loss $\mathcal{L}_{Teacher}$;
7:         Compute gradients $\nabla_\theta \mathcal{L}$;
8:         Update parameters $\theta \leftarrow \theta - \eta \nabla_\theta \mathcal{L}$;
9:     **end for**
10: **end for**

---

perspective, this means that intra-class distances should be smaller than inter-class distances. By introducing intra-class loss, the representation in the teacher model becomes more enriched. This ensures that the distilled knowledge transferred to the student model is not merely a replication of the real labels but includes deeper, class-specific insights.

### 3.3 MARGIN-BASED INTRA-CLASS CONTRASTIVE DISTILLATION

The teacher model trained with loss 5 may encounter several issues. First, there is a partial conflict between the cross-entropy loss and the intra-class loss, as the former encourages representations of samples within the same class to be more similar. This makes the model highly sensitive to the hyperparameter $\lambda$, potentially leading to model instability. Second, for samples that the teacher cannot correctly classify, increasing the intra-class distance is not reasonable. It is not desired intra-class distances exceed inter-class distances. Third, at the beginning of training, the model should focus on learning the differences between classes, and the presence of intra-class loss can slow down convergence.

**Definition 1** *Let $p^i$ denote the i-th degree. For each $x \in \mathcal{X}$ with the ground-truth label $y$, the margin is defined as*

$$\rho_x = p_x^y - \max_{i \in \mathcal{Y} \setminus \{y\}} p_x^i. \tag{6}$$

*Then, we have $-1 \leq \rho_x \leq 1$. A higher value of $\rho_x$ implies a higher proportion of the ground-truth label.*

We aim to select specific samples for intra-class contrastive learning through the use of margin. To achieve this, we simply choose anchor samples whose margin exceeds a specific threshold when calculating the intra-class contrastive loss.

$$\mathcal{L}_{Intra} = \begin{cases} -\log \frac{\exp(f(x)^\top f(x^+))}{\exp(f(x)^\top f(x^+)) + \sum_{k=1}^m \exp(f(x)^\top f(x_k^-))} & \text{if } \rho_x > \delta \\ 0 & \text{otherwise} \end{cases}, \tag{7}$$

where $\delta > 0$ is the threshold. Incorporating a margin threshold, $\delta$, into the intra-class contrastive distillation process strategically filters the anchor samples that contribute to the loss calculation. This approach focuses on strengthening the representations of those samples which are already well classified. Besides, the teacher selectively improves the feature embeddings only for samples where the confidence margin exceeds this value, effectively ignoring those where the model's certainty is low. This results in a more stable training process as it prevents the intra-class loss from overwhelming the model with conflicting gradients from poorly classified examples. Furthermore, it ensures that the learning process is not only focused on distinguishing between classes but also on refining the model's understanding within well-understood categories, thereby enhancing the overall effectiveness of the distillation process.

In practice, we observed that due to the constraints of GPU memory capacity, the number of negative samples per class in each batch is relatively small, with even fewer meeting the threshold. This

significantly impacts the performance of the intra-class loss. To address this issue, we adopted a pipeline-based approach: samples that meet the threshold are cached in a pipeline corresponding to their class. Once the number of samples in the pipeline is sufficient, we compute the intra-class loss and then clear the pipeline. This method not only substantially reduces memory consumption but also enhances the stability of intra-class contrastive learning.

In this section, we introduced the intra-Class Contrastive loss and discussed how it assists in generating soft labels by the teacher model. Addressing potential issues with the proposed intra-Class Contrastive Distillation, we have incorporated the concept of margin to improve the intra-Class Contrastive loss. The complete algorithm for the teacher model can be found in Algorithm 1.

## 4 THEORETICAL ANALYSIS

### 4.1 FORMULATION

In this section, we analyze the representations learned by the teacher model from the perspective of clustering. Our focus here is not on the model's classification performance, but rather on the distances within and between classes. Thus, in this section, the teacher model works as a feature embedding function $\varphi : \mathcal{X} \to \mathbb{R}^d (d > c)$, which transforms the data point from the $m$-dimensional sample space to the $d$-dimensional embedding space. We assume that $\varphi$ is normalized, such that $\|\varphi(x)\|_2 = 1$ for any $x \in \mathcal{X}$. Denote $\mathcal{H}$ as the hypothesis space of all embedding functions. For any sample $x$, the positive sample $x^+$ is an augmented version of $x$, while the negative samples $x^-$ are obtained through sampling.

In the previous sections, we proposed a method based on a tradeoff between cross-entropy loss and intra-class contrastive loss. In this section, we analyze the impact of the intra-class contrastive loss on the model. Since the teacher model performs supervised learning (with visible ground-truth labels), we employ cross-entropy loss to effectively learn inter-class differences. However, for the convenience of the following theoretical analysis, we substitute the loss function with a tradeoff between conventional inter-class contrastive loss and intra-class contrastive loss, i.e.,

$$
\mathcal{L}(\varphi) = \underbrace{-\log \frac{e^{\varphi(x)^T \varphi(x^+)}}{e^{\varphi(x)^T \varphi(x^+)} + \sum_{j=1}^n e^{\varphi(x)^T \varphi(x_j^-)}}}_{\mathcal{L}_{\text{Inter}}} + \lambda \cdot \underbrace{(-\log \frac{e^{\varphi(x)^T \varphi(x^+)}}{e^{\varphi(x)^T \varphi(x^+)} + \sum_{k=1}^m e^{\varphi(x)^T \varphi(x_k^-)}})}_{\mathcal{L}_{\text{Intra}}}.
$$

(8)

Here, $x^+$ is an augmented version of the sample $x$, serving as the positive sample, while $x_j^-$ and $x_k^-$ are negative samples. Notably, $x_j^- \notin C(x)$ in $\mathcal{L}_{\text{Inter}}$, whereas $x_k^- \in C(x)$ in $\mathcal{L}_{\text{Intra}}$. The inter-class loss $\mathcal{L}_{\text{Inter}}$ enables the model to learn the differences between different classes, ensuring that samples from different categories are well-separated in the feature space. Conversely, the intra-class loss $\mathcal{L}_{\text{Intra}}$ focuses on learning the differences within the same class, ensuring that samples of the same category are appropriately mutually distant in the feature space.

### 4.2 INTER-CLASS DISTANCES AND INTRA-CLASS DISTANCES

Before investigating our proposed method, we first consider a teacher model trained directly using cross-entropy loss.

**Proposition 1** *Consider $\lambda = 0$. If the model has perfect fitting capabilities, then the teacher model $f_T = \arg\min \mathcal{L}_{Teacher}$ in Eq.5 will induce soft labels that are identical to the ground-truth labels.*

This conclusion is evident. It shows that without some guidance for the teacher model, the generated soft labels may lose a significant amount of intra-class information. Now, we focus on the specifics of how our approach manages distance metrics within the embedding space. Essentially, the proposed inter-class contrastive loss and intra-class contrastive loss are designed to control inter-class distances and intra-class distances respectively. First, we define them as follows. Given the embedding $\varphi$, the inter-class distance is defined as

$$
d_{\text{Inter}} = \mathbb{E}_{x^- \notin C(x)} \left[ e^{\varphi(x)^T \varphi(x^-)} \right],
$$

and the intra-class distance is defined as

$$d_{\text{Intra}} = \mathbb{E}_{x^- \in C(x)} \left[ e^{\varphi(x)^T \varphi(x^-)} \right].$$

**Theorem 1** *As the two numbers of negative samples in 8 $n, m \to \infty$ and $n/m = K$, for a certain $\varphi$, we have $\frac{d_{Intra}}{d_{Inter}} = K \frac{e^{\mathcal{L}_{Intra}}}{e^{\mathcal{L}_{Inter}}}$.*

In fact, Theorem 1 connects the loss defined in 8 with inter-class and intra-class distances, illustrating that the proposed loss can effectively control both intra-class and inter-class distances. Furthermore, although the theorem requires that $n$ and $m$ approach infinity at the same rate, in practice, we can use the ratio of losses with finite samples as an approximation. Given the framework established by the previous sections and particularly by Theorem 1, we can explore the practical implications of these findings and further justify our method's approach. The two theorems essentially state that the relationship between the intra-class and inter-class distances can be quantitatively managed through the loss ratio.

Next, we will consider the impact of the intra-class contrastive loss on the model when optimizing the objective function

$$\min_{\varphi \in \mathcal{H}} \left\{ \mathcal{L}(\varphi) = \mathcal{L}_{\text{Inter}}(\varphi) + \lambda \mathcal{L}_{\text{Intra}}(\varphi) \right\}. \tag{9}$$

**Theorem 2** *Assume that $\varphi^* \in \arg\min_{\varphi \in \mathcal{H}} \mathcal{L}(\varphi)$, and then we have*

$$\frac{1}{C_0 \cdot \lambda + C_1} \leq \frac{\mathcal{L}_{Intra}}{\mathcal{L}_{Inter}} \leq C_2 \cdot \frac{1}{\lambda} + C_3, \tag{10}$$

*where $C_0, C_1, C_2, C_3$ are positive constants about $m$ and $n$.*

This theorem provides bounds for the ratio of intra-class and inter-class losses when optimizing the objective function $\mathcal{L}(\varphi)$. It shows how the balance parameter $\lambda$ affects the ratio between $\mathcal{L}_{\text{Inter}}$ and $\mathcal{L}_{\text{Intra}}$. The lower and upper bounds of this ratio are inversely related to $\lambda$. Thus, by adjusting $\lambda$, one can effectively control the balance between intra-class and inter-class optimization. A larger $\lambda$ leads to tighter intra-class samples, while a smaller $\lambda$ promotes better separation between different classes.

These theoretical results provide a solid justification for our approach by demonstrating how the contrastive loss, when modulated with a carefully chosen balance parameter $\lambda$, effectively manages the trade-off between intra-class compactness and inter-class separation. Moreover, the bounds in Theorem 2 offer assurance for both the control of intra-class diversity and the stability of the training process.

**Rationalizing the Loss Ratio** The control over the loss ratio enables a strategic balance between how closely the model learns to identify differences within the same class compared to those between different classes. This theoretical result is important as optimizing intra-class contrastive loss can improve intra-class distances. By optimizing both $\mathcal{L}_{\text{Intra}}$ and $\mathcal{L}_{\text{Inter}}$, the model learns to enhance discriminative ability not just at the inter-class level but also within the classes themselves.

**The choice of $\lambda$** From a practical standpoint, Theorem 2 illustrates the tradeoff between the two losses with respect to $\lambda$. In conjunction with Theorem 1, it allows us to adjust $\lambda$ to achieve an optimal balance between intra-class and inter-class distances. By choosing an appropriate value of $\lambda$, it is ensured that the model does not bias too much towards distinguishing only between classes and neglecting the variance within the classes, or vice versa. This balance is particularly beneficial in applications where subtle intra-class variations are as significant as the differences between classes.

In conclusion, the theoretical insights provided by the analysis not only bolster the validity of using a margin-based intra-class contrastive distillation approach but also highlight the importance of carefully considering the balance of loss components to achieve the best learning outcomes. As we move forward, these principles can guide the development of more sophisticated models that are tuned to the nuances of specific tasks and data characteristics.

Table 1: Results on CIFAR-100 and Tiny ImageNet, Homogenous Architecture. Top-1 accuracy is adopted as the evaluation criterion. All experiments are repeated 5 times, and the table presents the final mean of the results.

| Method | | CIFAR-100 | | | Tiny ImageNet | | |
|---|---|---|---|---|---|---|---|
| | Teacher | Resnet50 | WRN-40-2 | VGG13 | Resnet50 | WRN-40-2 | VGG13 |
| | | 78.31 | 76.89 | 74.40 | 59.48 | 59.48 | 55.91 |
| | Student | Resnet34 | WRN-16-2 | VGG8 | Resnet34 | WRN-16-2 | VGG8 |
| | | 78.19 | 76.4 | 73.80 | 58.56 | 58.68 | 53.80 |
| KD | | 78.38 | 76.6 | 73.99 | 59.47 | 59.17 | 53.68 |
| FitNet | | 75.20 | 75.03 | 72.33 | 56.43 | 58.09 | 50.29 |
| RKD | | 78.65 | 78.30 | 74.73 | 59.33 | 60.03 | 53.59 |
| CRD | | 78.57 | 78.69 | 74.48 | 58.89 | 59.19 | 53.69 |
| OFD | | 76.64 | 76.88 | 72.64 | 56.26 | 55.74 | 50.32 |
| ReviewKD | | 77.55 | 77.94 | 73.24 | 58.07 | 58.79 | 53.24 |
| VID | | 77.43 | 77.63 | 73.51 | 57.69 | 57.69 | 53.37 |
| MLLD | | 79.09 | 79.50 | 75.07 | 59.98 | 60.65 | 54.77 |
| Ours | | 79.10 | 79.09 | 74.96 | 59.64 | 60.16 | 54.96 |
| Ours+RKD | | **79.39** | **79.53** | **75.70** | **60.02** | **60.79** | **55.24** |

## 5 EXPERIMENTS

In this section, we evaluate the performance of the proposed Margin-Based Intra-Class Contrastive Distillation algorithm on image classification datasets.

### 5.1 DATASETS AND SETTINGS

**Datasets** We take three benchmark datasets: CIFAR-10, CIFAR-100 Krizhevsky (2009) and Tiny ImageNet. CIFAR-10 has 10 categories while CIFAR-100 has 100 categories. Both datasets consist of 50,000 training samples and 10,000 test samples, with an image resolution of 32x32. Tiny ImageNet Le & Yang (2015) contains 100000 images of 200 classes (500 for each class) downsized to 64×64 colored images. Each class has 500 training images, 50 validation images and 50 test images. Owing to space limitations, the experimental results for CIFAR-10 are provided in the appendix A.3.

**Setting** We compared two common settings in knowledge distillation: (1) Homogeneous architecture, where the teacher and student models share the same architecture, and (2) Heterogeneous architecture, where the teacher and student models have different architectures. Additionally, for each setting, we conducted experiments for various neural network architectures. The experimental results presented are the averages from 5 repeated trials. Owing to the limitations of page width, we have presented only the mean values without the standard deviation.

**Implementation Details** We set batchsize as 256 and the base learning rate as 0.5 for the teacher model, and set batchsize as 128 and the base learning rate as 0.05 for the student model. We adapt multi-step learning rate decay strategy. Both the teacher and student models are trained for 90 epochs, with the learning rate being reduced three times at epochs [30, 60]. When training the student model, the weight of the teacher's soft labels is set to 0.9. Stochastic Gradient Descent (SGD) is used as the optimizer for experiments. The weight $\lambda$ was set in the range of 0.01 to 0.03.

**Baselines** To validate the effectiveness of our proposed method, we conducted experiments comparing it with the vanilla KD Hinton et al. (2015) and seven benchmark methods: FitNet Romero et al. (2014), RKD Park et al. (2019), OFD Heo et al. (2019), ReviewKD Chen et al. (2021b), VID Ahn

Table 2: Results on CIFAR-100 and Tiny ImageNet, Heterogeneous Architecture. Top-1 accuracy is adopted as the evaluation criterion. All experiments are repeated 5 times, and the table presents the final mean of the results.

| Method | | CIFAR-100 | | | Tiny ImageNet | | |
|---|---|---|---|---|---|---|---|
| | Teacher | Resnet50 | VGG13 | WRN-40-2 | Resnet50 | VGG13 | WRN-40-2 |
| | | 78.31 | 74.40 | 76.89 | 58.77 | 55.91 | 59.48 |
| | Student | MobileNet | MobileNet | ShuffleV2 | MobileNet | MobileNet | ShuffleV2 |
| | | 65.18 | 65.18 | 69.23 | 40.29 | 40.29 | 44.12 |
| KD | | 66.22 | 65.82 | 70.16 | 41.82 | 44.68 | 46.81 |
| FitNet | | 67.52 | 63.51 | 65.06 | 43.69 | 40.64 | 37.04 |
| RKD | | 65.41 | 66.51 | 72.11 | 45.16 | 45.51 | 49.08 |
| ReviewKD | | 67.59 | 65.67 | 72.42 | 44.83 | 44.83 | 49.10 |
| VID | | 67.88 | 66.18 | 72.71 | 44.76 | 43.18 | 48.98 |
| MLLD | | 68.99 | 65.58 | 65.89 | 41.00 | 43.20 | 48.24 |
| Ours | | 68.72 | 66.44 | 72.00 | 45.32 | 46.01 | 48.64 |
| Ours+RKD | | **69.03** | **66.79** | **72.86** | **45.54** | **47.17** | **49.71** |

et al. (2019) and MLLD Jin et al. (2023). These methods provide a comprehensive backdrop against which the performance of our approach can be measured.

Unlike many traditional distillation methods that are designed to align the student model with the teacher model, our proposed algorithm specifically targets the teacher model to enrich the information contained in soft labels. Therefore, our approach can be integrated with many existing distillation techniques. In all experiments, we compare our Margin-Based Intra-Class Contrastive Distillation (ours) with other distillation algorithms. We also combined our algorithm with the classical RKD Park et al. (2019) (ours+RKD), which can achieve better performance.

Table 5 shows the results of the homogenous case, and Table 5.1 shows the results of the heterogeneous case. Our method achieves promising results in both cases. It also demonstrates that intra-class contrastive loss facilitates the learning of better soft labels, thereby enhancing the teaching capability of the teacher model and improving distillation effectiveness. Additionally, our approach can be integrated with many existing distillation algorithms, further enhancing the performance.

## 6 CONCLUSION

In this work, we introduced the Margin-Based Intra-Class Contrastive Distillation approach, which integrates intra-class contrastive learning with traditional knowledge distillation. This method not only enriches the soft labels with nuanced intra-class variations but also maintains a balance between inter-class and intra-class distinctions, which is vital for the robust generalization of the student model. Our method significantly enhances the performance of the student model by leveraging enriched soft labels, demonstrating superior results across standard image classification datasets. Both the integration of margin loss and the design of pipeline ensure the stability and efficiency of the learning process, addressing potential issues related to convergence and model training dynamics. The theoretical results also confirm the feasibility of our method and highlight the role of the parameter $\lambda$ in balancing the tradeoff. Overall, our approach provides a compelling framework for effectively utilizing the soft labels in knowledge distillation, paving the way for future innovations in model compression and efficient learning strategies.

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

## A APPENDIX / SUPPLEMENTAL MATERIAL

### A.1 PROOF OF THEOREM 1

By the definition of inter-class distance and intra-class distance, given a certain $\varphi$, we have

$$
\begin{aligned}
\frac{d_{\text{Intra}}}{d_{\text{Inter}}} &= \frac{\mathbb{E}_{x^- \notin C(x)}\left[e^{\varphi(x)^T \varphi(x^-)}\right]}{\mathbb{E}_{x^- \in C(x)}\left[e^{\varphi(x)^T \varphi(x^-)}\right]} \\
&= \lim_{\substack{m,n \to \infty \\ \frac{n}{m}=K}} \frac{\frac{1}{m}\sum_{i=1}^{m} e^{\varphi(x)^T \varphi(x^-)}}{\frac{1}{n}\sum_{j=1}^{n} e^{\varphi(x)^T \varphi(x^-)}} \\
&= K \lim_{\substack{m,n \to \infty \\ \frac{n}{m}=K}} \frac{\sum_{i=1}^{m} e^{\varphi(x)^T \varphi(x^-)}}{\sum_{j=1}^{n} e^{\varphi(x)^T \varphi(x^-)}}.
\end{aligned}
\tag{11}
$$

On the other side, according to the definitions of $\mathcal{L}_{\text{Inter}}$ and $\mathcal{L}_{\text{Intra}}$, there exists a relationship

$$
\frac{\sum_{i=1}^{m} e^{\varphi(x)^T \varphi(x^-)}}{\sum_{i=1}^{n} e^{\varphi(x)^T \varphi(x^-)}} = \frac{e^{\mathcal{L}_{\text{Intra}}} - 1}{e^{\mathcal{L}_{\text{Inter}}} - 1}.
\tag{12}
$$

Then, when $n, m \to \infty$ and $n/m = K$, the constant 1 can be ignored, and then we obtain

$$
\frac{d_{\text{Intra}}}{d_{\text{Inter}}} = K \frac{e^{\mathcal{L}_{\text{Intra}}}}{e^{\mathcal{L}_{\text{Inter}}}}.
\tag{13}
$$

### A.2 PROOF OF THEOREM 2

Recall that the total loss $\mathcal{L}(\varphi) = \mathcal{L}_{Inter}(\varphi) + \lambda \mathcal{L}_{Intra}(\varphi)$. Define

$$
\mathcal{L}_{Inter}^0(\varphi) = \mathcal{L}_{Inter}(\varphi) - \min_{\varphi \in \mathcal{H}} \mathcal{L}_{Inter}(\varphi)
\tag{14}
$$

$$
\mathcal{L}_{Intra}^0(\varphi) = \mathcal{L}_{Intra}(\varphi) - \min_{\varphi \in \mathcal{H}} \mathcal{L}_{Intra}(\varphi)
\tag{15}
$$

$$
\mathcal{L}^0(\varphi) = \mathcal{L}_{Inter}^0 + \lambda \mathcal{L}_{Intra}^0.
\tag{16}
$$

In fact, since $\mathcal{L}_{Inter}$ and $\mathcal{L}_{Intra}$ are both non-negative, we can approximate the lower bound with an arbitrarily small $\varepsilon$ gap. For simplicity, we assume that the lower bound can be achieved, and let $\varphi_1 = \arg\min_{\varphi \in \mathcal{H}} \mathcal{L}_{Inter}$ and $\varphi_2 = \arg\min_{\varphi \in \mathcal{H}} \mathcal{L}_{Intra}$. Thus, it is clear that $\mathcal{L}_{\text{Inter}}^0(\varphi_1) = 0$ and $\mathcal{L}_{\text{Intra}}^0(\varphi_2) = 0$ hold.

Next, we provide a precise estimate for the lower bound of $\min_{\varphi \in \mathcal{H}} \mathcal{L}_{Inter}(\varphi)$ and $\min_{\varphi \in \mathcal{H}} \mathcal{L}_{Intra}(\varphi)$.

$$
\begin{aligned}
\mathcal{L}_{Inter}(\varphi) &= \log\left(1 + \frac{\sum_{j=1}^{n} e^{\varphi(x)^T \varphi(x_j^-)}}{e^{\varphi(x)^T \varphi(x^+)}}\right) \\
&\geq \log\left(1 + \frac{\sum_{j=1}^{n} e^{-1}}{e^1}\right) \\
&= \log\left(1 + n e^{-2}\right).
\end{aligned}
\tag{17}
$$

The equality is satisfied when $\varphi(x)^T \varphi(x^+) = 1$ and $\varphi(x)^T \varphi(x_j^-) = -1$, for $j = 1, 2, \ldots, n$. On the other side,

$$
\begin{aligned}
\mathcal{L}_{Intra}(\varphi) &= \log\left(1 + \frac{\sum_{i=1}^{m} e^{\varphi(x)^T \varphi(x_i^-)}}{e^{\varphi(x)^T \varphi(x^+)}}\right) \\
&\geq \log\left(1 + \frac{\sum_{i=1}^{m} e^{-1}}{e^1}\right) \\
&= \log\left(1 + m e^{-2}\right).
\end{aligned}
\tag{18}
$$

The equality is satisfied when $\varphi(x)^T\varphi(x^+) = 1$ and $\varphi(x)^T\varphi(x_i^-) = -1$, for $i = 1, 2, \ldots, m$.

Assume that $\varphi^* \in \arg\min_{\varphi \in \mathcal{H}} \mathcal{L}(\varphi)$. Thus, it is evident that $\varphi^* \in \arg\min_{\varphi \in \mathcal{H}} \mathcal{L}^0(\varphi)$. Since that $\mathcal{L}^0_{\text{Inter}}(\varphi_1) = 0$, by the optimality of $\varphi^*$, we have that

$$
\begin{aligned}
\mathcal{L}^0_{Inter}(\varphi^*) + \lambda\mathcal{L}^0_{Intra}(\varphi^*) &\leq \mathcal{L}^0_{Inter}(\varphi_1) + \lambda\mathcal{L}^0_{Intra}(\varphi_1) \\
&= \lambda\mathcal{L}^0_{Intra}(\varphi_1).
\end{aligned}
\tag{19}
$$

Upon expanding the equation, we obtain

$$
\begin{aligned}
&\mathcal{L}_{Inter}(\varphi^*) - \min_{\varphi \in \mathcal{H}} \mathcal{L}_{Inter}(\varphi) + \lambda\mathcal{L}_{Intra}(\varphi^*) - \lambda\min_{\varphi \in \mathcal{H}} \mathcal{L}_{Intra}(\varphi) \\
&\leq \lambda\mathcal{L}_{Intra}(\varphi_1) - \lambda\min_{\varphi \in \mathcal{H}} \mathcal{L}_{Intra}(\varphi).
\end{aligned}
\tag{20}
$$

Rearrange this equation, and thus

$$
\begin{aligned}
\frac{\mathcal{L}_{Inter}(\varphi^*)}{\mathcal{L}_{Intra}(\varphi^*)} &\leq \frac{\lambda\mathcal{L}_{Intra}(\varphi_1) - \lambda\mathcal{L}_{Intra}(\varphi^*) + \min_{\varphi \in \mathcal{H}} \mathcal{L}_{Inter}(\varphi)}{\mathcal{L}_{Intra}(\varphi^*)} \\
&= \lambda\frac{\mathcal{L}_{Intra}(\varphi_1)}{\mathcal{L}_{Intra}(\varphi^*)} + \frac{\min_{\varphi \in \mathcal{H}} \mathcal{L}_{Inter}(\varphi)}{\mathcal{L}_{Intra}(\varphi^*)} - \lambda \\
&= \lambda\frac{\log\left(1 + \frac{\sum_{i=1}^m e^{\varphi_1(x)^T\varphi_1(x_i^-)}}{e^{\varphi_1(x)^T\varphi_1(x^+)}}\right)}{\log\left(1 + \frac{\sum_{i=1}^m e^{\varphi^*(x)^T\varphi^*(x_i^-)}}{e^{\varphi^*(x)^T\varphi^*(x^+)}}\right)} + \frac{\log\left(1 + ne^{-2}\right)}{\log\left(1 + \frac{\sum_{i=1}^m e^{\varphi^*(x)^T\varphi^*(x_i^-)}}{e^{\varphi^*(x)^T\varphi^*(x^+)}}\right)} - \lambda \\
&\leq \lambda\frac{\log\left(1 + \frac{me}{e^{-1}}\right)}{\log\left(1 + \frac{me^{-1}}{e}\right)} + \frac{\log\left(1 + ne^{-2}\right)}{\log\left(1 + \frac{me^{-1}}{e}\right)} - \lambda \\
&= \left(\frac{\log\left(1 + me^2\right)}{\log\left(1 + me^{-2}\right)} - 1\right)\lambda + \frac{\log\left(1 + ne^{-2}\right)}{\log\left(1 + me^{-2}\right)} \\
&= C_0 \cdot \lambda + C_1,
\end{aligned}
\tag{21}
$$

where $C_0, C_1 > 0$.

Since both sides are non-negative, taking the reciprocal yields

$$
\frac{\mathcal{L}_{Intra}(\varphi^*)}{\mathcal{L}_{Inter}(\varphi^*)} \geq \frac{1}{C_0 \cdot \lambda + C_1}.
\tag{22}
$$

Similarly, for $\varphi_2$ satisfying $\mathcal{L}^0_{\text{Intra}}(\varphi_2) = 0$, by the optimality of $\varphi^*$, we have that

$$
\begin{aligned}
\mathcal{L}^0_{Inter}(\varphi^*) + \lambda\mathcal{L}^0_{Intra}(\varphi^*) &\leq \mathcal{L}^0_{Inter}(\varphi_2) + \lambda\mathcal{L}^0_{Intra}(\varphi_2) \\
&= \mathcal{L}^0_{Inter}(\varphi_2).
\end{aligned}
\tag{23}
$$

Upon expanding the equation, we obtain

$$
\begin{aligned}
&\mathcal{L}_{Inter}(\varphi^*) - \min_{\varphi \in \mathcal{H}} \mathcal{L}_{Inter}(\varphi) + \lambda\mathcal{L}_{Intra}(\varphi^*) - \lambda\min_{\varphi \in \mathcal{H}} \mathcal{L}_{Intra}(\varphi) \\
&\leq \mathcal{L}_{Inter}(\varphi_2) - \min_{\varphi \in \mathcal{H}} \mathcal{L}_{Inter}(\varphi).
\end{aligned}
\tag{24}
$$

Table 3: Results on CIFAR-10, Homogenous and Heterogeneous Architectures. Top-1 accuracy is adopted as the evaluation criterion. All experiments are repeated 5 times, and the table presents the final mean of the results.

| Method | | CIFAR-10 Homogenous | | | CIFAR-10 Heterogeneous | | |
|---|---|---|---|---|---|---|---|
| | Teacher | Resnet50 | WRN-40-2 | VGG13 | Resnet50 | WRN-40-2 | VGG13 |
| | | 94.08 | 94.85 | 91.54 | 94.08 | 94.85 | 91.54 |
| | Student | Resnet34 | WRN-16-2 | VGG8 | MobileNetV2 | ShuffleV2 | MobileNetV2 |
| | | 93.16 | 93.22 | 91.04 | 83.46 | 87.09 | 83.46 |
| KD | | 93.92 | 93.82 | 91.96 | 84.72 | 88.64 | 78.83 |
| FitNet | | 93.53 | 93.74 | 91.38 | 86.39 | 85.48 | 82.70 |
| RKD | | 94.13 | 94.05 | 92.22 | 86.26 | 89.41 | 85.21 |
| CRD | | 90.97 | 90.70 | 89.31 | - | - | - |
| OFD | | 94.08 | 94.09 | 92.01 | - | - | - |
| ReviewKD | | 94.04 | 94.02 | 92.18 | 86.09 | 90.86 | 83.97 |
| VID | | 94.02 | 93.99 | 91.87 | 86.49 | 90.31 | 86.23 |
| MLLD | | 92.32 | 92.18 | 90.28 | 85.78 | 87.33 | 84.12 |
| Ours | | 94.17 | 94.23 | 92.33 | 86.51 | 90.96 | 85.98 |
| Ours+RKD | | **94.36** | **94.29** | **92.25** | **86.81** | **91.91** | **86.59** |

Rearrange this equation, and thus

$$
\begin{aligned}
\frac{\mathcal{L}_{Intra}(\varphi^*)}{\mathcal{L}_{Inter}(\varphi^*)} &\leq \frac{\mathcal{L}_{Inter}(\varphi_2) + \lambda \min_{\varphi \in \mathcal{H}} \mathcal{L}_{Intra}(\varphi)}{\lambda \mathcal{L}_{Inter}(\varphi^*)} - \frac{1}{\lambda} \\
&= \frac{\log\left(1 + \frac{\sum_{j=1}^n e^{\varphi_2(x)^T \varphi_2(x_j^-)}}{e^{\varphi_2(x)^T \varphi_2(x^+)}}\right) + \lambda \log(1 + me^{-2})}{\lambda \log\left(1 + \frac{\sum_{j=1}^n e^{\varphi^*(x)^T \varphi^*(x_j^-)}}{e^{\varphi^*(x)^T \varphi^*(x^+)}}\right)} - \frac{1}{\lambda} \\
&\leq \frac{\log\left(1 + \frac{ne}{e^{-1}}\right)}{\lambda \log\left(1 + \frac{ne^{-1}}{e}\right)} - \frac{1}{\lambda} + \frac{\log(1 + me^{-2})}{\log\left(1 + \frac{ne^{-1}}{e}\right)} \\
&= \left(\frac{\log\left(1 + ne^2\right)}{\log\left(1 + ne^{-2}\right)} - 1\right)\frac{1}{\lambda} + \frac{\log(1 + me^{-2})}{\log\left(1 + ne^{-2}\right)} \\
&= C_2 \cdot \frac{1}{\lambda} + C_3,
\end{aligned}
\tag{25}
$$

where $C_2, C_3 > 0$.

Combining Eq.22 and Eq.25 completes the proof.

## A.3 OTHER EXPRIMENTS

Table A.3 shows the results on CIFAR-10.

