# OpenReview forum: "Enriching Knowledge Distillation with Intra-Class Contrastive Learning"
_ICLR.cc/2025/Conference — ICLR 2025 Conference Withdrawn Submission_

### Official Review · Reviewer_19e5 · 2024-10-15

**Soundness:** 2
**Presentation:** 3
**Contribution:** 2
**Rating:** 3
**Confidence:** 5

**Summary:**

In this manuscript, the authors present a knowledge distillation method named Margin-Based Intra-Class Contrastive Distillation. The proposed method incorporates an intra-class contrastive loss to enrich the soft labels generated by the teacher model during training, which encourages the teacher to learn diverse representations within each class, thereby providing a richer knowledge transfer to the student model.

**Strengths:**

1. The manuscript provides a theoretical analysis to demonstrate the impact of intra-class contrastive loss on the model's representation learning as well as the quantitative relationship between intra-class and inter-class distances.
2. The margin loss is proposed to improve training stability and convergence speed which may cope with the intra-class contrastive loss.
3. Experimental results show good performance against comparison methods.
4. This manuscript is well-written and clearly organized, where the introduction section effectively motivates the knowledge distillation problem and highlights the key contributions of this manuscript.

**Weaknesses:**

1. The novelty of this manuscript is minor. There are works concerned with intra-class distance, such as "CKD: Contrastive Knowledge Distillation from A Sample-Wise Perspective". CKD employs intra-sample similarities and inter-sample dissimilarities and formulates these constraints into a contrastive learning framework. The authors should claim the difference against CKD and compare with it.
2. The theoretical analysis only concentrates on relationships between intra-class and inter-class distances for the selection of parameter \lambda. However, it may lack a deeper exploration of how intra-class diversity truly affects the performance of the student model.
3. The experimental results are not convincing. First, the comparison methods should include baselines that specifically address intra-class diversity. Second, the evaluation datasets are limited, previous methods conduct experiments on CIFAR-100, MS-COCO, and ImageNet-1K for image classification and object detection. Third, the ablation study is not provided to prove the effectiveness of intra-class and inter-class distances. Moreover, more network architectures should be used for comparison.
4. Missing key references, such as "PromptKD: Unsupervised Prompt Distillation for Vision-Language Models" and "CKD: Contrastive Knowledge Distillation from A Sample-Wise Perspective".
5. The format of some references is not correct.

**Questions:**

Suggestions:
1. more comparisons with recent knowledge distillation methods, especially those that use contrastive learning.
2. provide ablation studies to assess the contributions of margin loss and intra-class losses.
3. more evaluation datasets and network architectures.

---

### Official Review · Reviewer_pFZt · 2024-10-31

**Soundness:** 1
**Presentation:** 1
**Contribution:** 2
**Rating:** 3
**Confidence:** 3

**Summary:**

This paper introduces an intra-contrastive loss that pulls an augmented sample closer to its original counterpart while pushing other samples from the same class farther apart in the teacher model's embedding space. The teacher model is trained using a combination of intra-contrastive loss and cross-entropy loss, balanced by a weighting factor, $\lambda$. This approach aims to distill richer intra-class knowledge and inter-class knowledge from the teacher model to the student model. However, the authors identify three problems associated with this training loss that lead to instability in the training phase and slow convergence. To address these issues, the paper proposes a margin loss that applies intra-contrastive loss only to samples whose predicted probability for the ground-truth label exceeds a threshold during training. To validate the proposed loss, they define inter-class and intra-class distance metrics and demonstrate that the proposed intra-class and inter-class losses correspond to their respective metrics. Additionally, they prove that $\lambda$ can adjust the trade-off between intra-class and inter-class separation.

**Strengths:**

Based on the theoretical proof, this paper implements the trade-offs between intra-class and inter-class separation. These theoretical results can inspire the ICLR community.

**Weaknesses:**

In the theoretical sections, to prove Theorem 1 and Theorem 2, this paper assumes that the cross-entropy loss is equivalent to the inter-class contrastive loss. However, they doesn't demonstrate  that these two loss functions serve a similar purpose, either theoretically or experimentally. Could you explain, if your assumption holds, why you did not use the inter-class contrastive loss directly for training the teacher model instead of the cross-entropy loss? For Theorem 1, the paper defines two distance metrics but does not demonstrate whether these metrics satisfy the four conditions required for a valid distance function (metric space).

Based on these theorems, this paper trains the teacher model to enrich intra-class information, whereas other methods simply utilize an already trained teacher model. However, the paper does not demonstrate the effectiveness of training the teacher model and distilling knowledge to the student, even at the expense of increased training cost. Specifically, the experiments do not report the results of the teacher model trained with their loss, nor do they compare the student model’s improved performance when combined with other KD methods (beyond RKD) against the additional parameters that need to be trained. Furthermore, could you provide experimental results for directly training the student model using  your proposed losses (CE and margin) along with knowledge distillation loss, rather than just distilling knowledge through the teacher model trained with those losses?

**Questions:**

1. Could you provide any theoretical or experimental demonstrations to support your assumption that cross-entropy loss is equivalent to inter-class contrastive loss?
2. If your assumption holds, could you explain why you did not directly use the inter-class contrastive loss for training the teacher model instead of cross-entropy loss?
3. Could you show that your two distance metrics (inter-class and intra-class distance) satisfy the four conditions required for a valid distance function (metric space)?
4. In section 3.1, you define the sample space($\mathcal{X}$), the label space($\mathcal{Y}$) and the hypothesis space($\mathcal{F} : \mathcal{X} \rightarrow \mathcal{Y}$) then refer to the classifier as $f \in \mathcal{F}$. However, shouldn't it be represented as $\mathcal{F} : \mathcal{X} \rightarrow \mathbb{R}^d(d>c)$, similar to the feature embedding function $\varphi$ described in Section 4.1?
5. Could you report  the results of the teacher model trained with your loss?
6. Could you compare the student model’s improved performance when combined with other KD methods (beyond RKD) against the additional parameters that need to be trained?
7. Could you provide experimental results for directly training the student model using  your proposed losses (CE and margin) along with knowledge distillation loss, rather than just distilling knowledge through the teacher model trained with those losses?
8.  Could you verify whether Sohn (2016a) and Oord et al. (2018) on page 2 indeed employ augmented samples as positive samples and other samples from the same class as negative samples?
9.  Could you report the test accuracy and test loss curves for at least one type of teacher model to illustrate instability and slow convergence during training?

Things to improve the paper that did not impact the score:
1.  Please follow the formatting instructions of ICLR regarding citations within the text, ensuring that \citep{} and \citet{} are used appropriately.
2.  Possible typo: In line 5 of the Contrastive Learning section on page 3, "Constrastive earning" should be corrected to "Contrastive learning."
3.  Please differentiate between scalar and vector forms in your mathematical notation.
4.  Please clarify whether $f$ indicates $f_t$ or $f_s$ in Equations (4) and (7).
5.  To my knowledge, existing methods utilize KL-divergence for knowledge distillation. Strictly speaking, While KL-divergence and cross-entropy loss may function similarly in knowledge distillation, they are mathematically different. Please clarify the mathematical differences in Equations (2) and (3).
6. What type of augmentation techniques did you use for positive samples in the intra-class loss?
7. what values did you set for $\delta$ in the margin loss?
8. Please unify the notation for referencing equations, such as whether to use "Equation 5," "Eq. 5," or simply "5."

---

### Official Review · Reviewer_562d · 2024-11-01

**Soundness:** 3
**Presentation:** 3
**Contribution:** 3
**Rating:** 5
**Confidence:** 5

**Summary:**

The paper proposes a contrastive learning method for training the teacher model in knowledge distillation, aiming to provide student models with richer intra-class information from the teacher model.

**Strengths:**

The paper observes that the teacher model trained with ground truth may lack intra-class information for students, which is interesting and meaningful.

The paper is well-written and easy to follow.

**Weaknesses:**

1) The paper claims that the proposed method improves training stability and convergence speed; however, there are no experiments demonstrating this advantage.

2) There is a lack of ablation studies on the threshold \theta and batch size (batch size is an important hyperparameter for contrastive learning).

3) The proposed method requires retraining the teacher models and lacks a comparison of the time required.

4) There is a lack of experiments on large datasets like ImageNet.

5) Without RKD, the proposed method underperforms compared to existing methods.

6) It would be better to compare to the dynamic temperature methods [1] and other state-of-the-art methods like [2].

**Questions:**

1) Why not use the common hyperparameter settings for KD? For example, for CIFAR-100, use 240 epochs with a learning rate of 0.05 and a batch size of 64 [3].

2) What teacher models were used for the baseline methods?

3) What is the performance of the proposed method when combined with other baseline methods?

4) The experiments lack CRD [3], which proposed contrastive learning for KD.

5) please address W1-4.

6) It would be better to have a comparison to a teacher trained with inter-class loss.

[1] Li, Z.; Li, X.; Yang, L.; Zhao, B.; Song, R.; Luo, L.; Li, J.; and Yang, J. 2023. Curriculum temperature for knowledge distillation. In Association for the Advancement of Artificial Intelligence (AAAI)

[2] Sun, S.; Ren, W.; Li, J.; Wang, R.; and Cao, X. 2024. Logit Standardization in Knowledge Distillation. In Proc. IEEE Conf. on Computer Vision and Pattern Recognition (CVPR).

[3] Tian, Y.; Krishnan, D.; and Isola, P. 2020. Contrastive representation distillation. Proc. Int. Conf. on Learning Representation (ICLR)

---

### Official Review · Reviewer_DsYQ · 2024-11-02

**Soundness:** 3
**Presentation:** 3
**Contribution:** 2
**Rating:** 3
**Confidence:** 5

**Summary:**

This paper proposes the Margin-Based Intra-Class Contrastive Distillation approach, which integrates intra-class contrastive learning with traditional knowledge distillation.

**Strengths:**

This paper is well-written and well-structured.

**Weaknesses:**

1. The idea of ​​enhancing knowledge distillation via contrastive learning is not innovative.
2. Based on the experimental results, the improvement of the proposed method is not significant.

**Questions:**

Here are some concerns that need to be addressed.
1. The idea of ​​enhancing knowledge distillation via contrastive learning is not innovative.
2. Based on the experimental results, the improvement of the proposed method is not significant.
3. Following the previous work, the author needs to add experimental analysis results on the ImageNet and MS COCO datasets.
4. Moreover, the authors need to add relevant analysis regarding the efficiency of the proposed method compared to the state-of-the-art baselines, including training time.

---

### Official Review · Reviewer_EoBB · 2024-11-03

**Soundness:** 3
**Presentation:** 3
**Contribution:** 2
**Rating:** 5
**Confidence:** 3

**Summary:**

The paper proposes a novel approach to knowledge distillation by incorporating intra-class contrastive learning to enrich the information contained in soft labels. The key contributions include:
- A new intra-class contrastive loss function that encourages appropriate separation between samples of the same class
- Integration of margin loss to improve training stability and convergence
- Theoretical analysis of the relationship between intra-class contrastive loss and feature distances
- Empirical validation on standard image classification benchmarks

**Strengths:**

- Strong theoretical foundation with formal proofs for the proposed method
- Novel perspective on enriching soft labels through intra-class information
- Practical implementation considerations (pipeline-based caching mechanism)
- Comprehensive empirical evaluation across multiple architectures and datasets
- Clear connection to existing literature and proper positioning of contributions

**Weaknesses:**

- Introducing intra-class contrastive learning and margin loss increases the complexity of the model, which may make training and tuning more difficult in practical applications with limited resources, especially in scenarios where overly complex models are not easily deployable. Provide a quantitative analysis of the increased computational complexity or memory requirements compared to standard knowledge distillation.
- Although margin loss helps improve training stability, in cases of small datasets or unbalanced samples, intra-class contrastive loss may still lead to training instability, affecting the model's convergence speed and performance.  Provide experimental results or analysis specifically addressing the performance on small or imbalanced datasets.
- Additionally, the experiments mainly focus on specific benchmark datasets, lacking extensive validation across different types of datasets, especially in fields such as natural language processing and time series analysis, where their applicability has not been fully assessed.
- Tuning weight parameters (such as α and λ) requires careful consideration, increasing complexity and potentially leading to inconsistent performance. Provide a sensitivity analysis or guidelines for tuning these parameters.
- Although the paper provides some theoretical analysis, the discussion on how intra-class contrastive loss specifically affects the model's learning mechanism is still insufficient, and further theoretical research will help to gain a deeper understanding of the principles of this method.
- Limited novelty: The core idea combines existing concepts (contrastive learning and margin-based approaches)
- Experimental analysis lacks ablation studies showing the individual impact of different components
- No discussion of computational overhead introduced by the additional loss terms
- Limited exploration of hyperparameter sensitivity, especially for λ and margin threshold δ
- Results on CIFAR-100 and Tiny ImageNet show only modest improvements over existing methods

**Questions:**

Although the paper mentions that margin loss improves training stability, if unstable phenomena are observed in experiments, it is recommended that the authors provide more details on training stability analysis and coping strategies, including experimental results under different hyperparameter settings. The experiments mainly focus on specific image classification datasets, and it is suggested to expand the scope of experiments to cover text, audio, or time series data to verify the generality and applicability of the method. In addition, it is recommended to provide more detailed parameter tuning guidelines to help researchers effectively select weight parameters (such as α and λ). Finally, it is suggested to delve into the theoretical basis of intra-class contrastive loss to fully understand its specific impact on the model's learning mechanisms.
1. How does the computational complexity compare to standard knowledge distillation?
2. What is the sensitivity of the method to the choice of λ and δ?
3. How does the pipeline-based caching mechanism affect training time?
4. Can you provide ablation studies showing the individual contribution of intra-class contrastive loss and margin loss?

---

### Note · Authors · 2024-11-17

I have read and agree with the venue's withdrawal policy on behalf of myself and my co-authors.